# Observations of Bell Inequality Violations with Causal Isolation between Source and Detectors

**DOI:** 10.3390/e24091230

**Published:** 2022-09-01

**Authors:** Marc Jean Jose Fleury

**Affiliations:** Independent Researcher, 3344 Peachtree Rd NE, Atlanta, GA 30326, USA; marcf999@gmail.com

**Keywords:** quantum entanglement, Bell inequality violations, measurement loophole, standing waves

## Abstract

We report the experimental observations of Bell inequality violations (BIV) in entangled photons causally separated by a rotating mirror. A Foucault mirror gating geometry is used to causally isolate the entangled photon source and detectors. We report an observed BIV of CHSH-S=2.30±0.07>2.00. This result rules out theories that explain correlations with traveling communication between source and detectors, including super-luminal and instantaneous communication.

## 1. Motivation and Hypothesis

### 1.1. On Bell Inequality Violations and Local Realism

It is commonly accepted that the Aspect [1] and Zeilinger [2] experiments convincingly closed the locality loophole for Bell inequality violations (BIV) experiments. They do so by causally isolating the acts of measurements at two distinct points called Alice and Bob. Modern BIV experiments close the locality loophole in increasingly convincing fashion. See [3] for a submarine fiber implementation, ref. [4] for an orbital station implementation, and [5] for measurement settings triggered by distant stars. The experiments all focus on isolating the measuring stations from one another: Alice and Bob are the ones causally separated, by land, sea, satellites and deep space. While the experiments undoubtedly achieve observer to observer isolation we will argue in Section 1.2 that these experiments do not isolate the emitting atoms from the observers. We will argue that in all these geometries, there is an ever-present line of sight between the polarizers and the source. We will argue that such influence of the measuring stations on the source can account for BIVs in a classic local realistic ways in Section 1.3. It follows that we might lose BIV by removing the line of sight between source and detection. A Foucault mirror design gates and effectively removes the line of sight between observers and emitters. With this gate in place, we perform a Clauser–Horne–Shimony–Holt (CHSH) measure of BIV in a 2-channel setup. In Section 3, we report on the observation of CHSH S=2.30>2. This rules out travelling influences between source and detectors, including super-luminal, as candidates for the Bell effect in Section 4.

### 1.2. On Experimental Locality: Observer-to-Observer vs. Observer-to-Emitting Atoms

We will now review the isolation experimentally achieved thus far in the current state of the art experiments. We will observe a static line of sight between emitting atoms and observing ones. Bell famously prescribed “randomly setting the polarizers during the flight of the photons” as a way to close the locality loophole. Modern experiments causally isolate the observers between themselves. In the Aspect and Weihs–Zeilinger experiments, we never set the direction of the polarization analyzers during the flight of photons. This is, in fact, mechanically impossible [6]. Instead, the polarizers are all preset, and we dynamically route the photons to a randomly chosen analyzer. This is achieved by randomly switching the path of the photons while in flight. We then claim physical analogy to “setting the polarizers”, but the polarizers were chosen more than they were set. The settings are, in fact, all preset; it is only the choice of the polarizers which is dynamic and random. To effect this random optical switch, the original experiments by Alain Aspect used acousto-optic modulators (AOMs) operating in the MHz domain [1]; the modern state-of-the-art experiments in the vein of Weihs and Zeilinger use electro-optic modulators (EOMs) operating in the GHz domain [2,7]. These AE/OM devices randomly switch the path of the photons and therefore randomly choose which one of the preset polarization analyzers will perform the actual measure. Both experiments are based on a similar implementation concept of the optical switches and polarizers: they preset all polarization analyzers upfront and then choose one by randomly modulating the index of refraction of some media, either acoustically or electrically. In the EOM experiments, the emitting atoms inside the barium borate (BBO) crystal have a permanent line of sight to all four polarizers with static settings. The EOMs modulate the phases, and the AOMs modulate amplitudes. The AOMs behave more like switching mirrors with a 4:1 reflection contrast. Therefore, a line of sight from the detectors to the emitting cascading atoms is also always present in the Aspect experiments. The presence of the AOM in the path does not remove this line of sight; it modulates the amplitudes of signals coming through it. State-of-the-art AOMs never realize a true zero in amplitude transmission; the residual signal is rather large. There is always a line of sight. Therefore, we conclude that the atoms in the sources of both geometries (SPCD/BBO and cascading atoms) are connected to all four static polarizers at all times. The emission of the entangled photons happens with full visibility of the polarizers at a fixed angle in both types of experiments. See Figure 1 for a schematic rendering of isolation achieved with these AE/OMs. This line of sight was hiding in plain sight, so to speak.

### 1.3. Hypothesis: On a Background Influence between Emitting Atoms and Polarizer Atoms

With regards to Bell’s theorem, such a background field would allow for Bell-type correlations in simple local realistic ways. Such a de-localized background field trivially breaks the isolation assumption of the theorem according to [8]. This is not in violation of, but rather in accord with, Bell’s theorem. We will now briefly review the literature for BIVs and show that there is little agreement as to what is causing the Bell correlations. For example, other modern local realist approaches hypothesize non-linear amplifications of the so-called zero-point field modes combined with a hypothetical threshold detection in avalanche photodetectors (APDs) [9,10]. Others, more generically, focus on Bohr contextuality pointing out that the measures always interfere with the measured. Both the photons in flight and the emitting source atoms are subject to dynamic influences [11,12,13]. In the view of contextuality, Bell inequalities may be derived using probabilistic models describing four random experiments performed in incompatible experimental contexts [14]. In this view, all speculations about a Bell “effect” are based on an incorrect causal interpretation of conditional probabilities [15]. Others argue the physical fact that Bell-type correlations emerge from classical electromagnetism [16]. Counting coincidences in a Bell game is, after all, measuring intensity at two different points. This intensity interferometer (as opposed to amplitude) is known as a Hanbury–Brown–Twiss (HBT) interferometer [17]. Others still within the HBT interferometer approach reconstruct the proper Bell intensity correlations with a classical EM local realist model [18]. They hypothesize circularly polarized single photons coming out of the SPDC process in the BBO crystals. By definition of circular polarization, this introduces a quarter-wave phase delay between the horizontal and vertical modes of the BBO crystals. This is in clear opposition to the understanding of BBO crystal emissions in [6]. There, the horizontal and vertical modes are understood to be in phase, which results in a “quantum superposition” of the linear modes. This is called a singlet or triplet state, depending on the sign of the delay. In fact, as reported in Weihs’ Ph.D. thesis (German only), one fine-tunes the exit phases of the BBO modes so as to be in phase, a “sine-qua-non” condition of the observation of BIVs and a big part of the experimental approach. This results in a ‘quantum superposition’ by erasing the which-path information contained in the phases. Finally, and as if to further confuse everyone in the ivory tower of B(e)aBel(l) [19], some have recently reported observations of Bell-type correlations using *uncorrelated* laser sources [20].

It stands to reason that if field communication is responsible for the Bell effect, then when we remove said field, we can anticipate a loss of the Bell effect (no line of sight? no BIV!). It is simple to test this background influence hypothesis: we introduce a Foucault mirror designed to disrupt the EM line of sight between emitting atoms and polarizers. With this rotating mirror, we remove any EM background field between observers and emitters. How does this affect a Bell–CHSH measure?

## 2. Experimental Setup

### A 2-Channel Photon Entanglement Measure Using a Rotating Mirror

To test this background field hypothesis, we introduced a spinning Foucault mirror in a classic 2-channel Bell experiment. See Figure 2. This was a simple table-top 2-channel experiment showing a Bell–CHSH inequality violation with static settings of the polarizers, as opposed to the more evolved 4-channel detection loophole experiments using dynamic choice (and static setting) of the measures reviewed above. We removed the line of sight by introducing a rotating Foucault mirror, inspired from the eponymous 1850 measure of the speed of light. This connected the components only for a brief amount of time, and the source was optically isolated from the polarizers for the vast majority of the time. Contrast this to the AE/OM experiments, where the source is continuously connected to the polarizers. Here, we only focused on the fiber communication. We did not care about physical line-of-flight isolation; in fact, everything fit on a table-top optical bread board. This setup did not close the locality loophole in the usual spatial sense, since the detectors were physically close to each other. We only considered the optical isolation achieved by the fiber. This type of strict causal isolation of the source from the polarizers via a fiber and mirror gate is new and therefore, to the best of our knowledge, an experimental first.

An entanglement source commercially available from quTools, GmbH, allows for a classic 2-channel singlet experiment showing Bell–CHSH violations. A singlet state of entangled photons was prepared in a superposition of horizontally and vertically polarized photons. Both photons were sent down 200 m of single-mode fiber into a single rotating mirror (1000 Hz, 34 facets) purchased from Cambridge Technologies (model SA34 P/N:1-1-3304-001-00). The geometrical arc described by the beams reflected off each of the rotating mirror facets covered the slits over Alice and Bob. See Figure 2. The result was a gate: only when the rotating beams cover the openings over the slits were the gates opened and the components connected. The beams were then polarized by going though static linear polarizers with axes that were computer controlled. Single photon avalanche photodetectors counted the incoming filtered photons, giving us the single counts in channel A and B. A quTools-provided coincidence detector circuit counting simultaneous detection in the A and B channels completed the setup.


**Time of Aperture:**


We can adjust the time this window stays open and during which the paths are connected. The time of aperture is determined by the aperture width of the slits $A=10^{-3}$ m, the rotational velocity $w=10^{3}$ Hz, and the physical distance between the slits and the mirror on our bread board $r=34\times 10^{-2}$ m:(1)$$T_{on}=\frac{A}{2\pi rw}=4.7\times 10^{-7}\;s$$

Since we used the same mirror for both paths, the shutter effects at both slits were in phase. The slits were illuminated at the same time, and we thus detect coincidences in the APDs. Note that two different mirrors would not be in phase, and we would not detect coincidences in the counters because the APDs would not be iluminated at the same time, or at least within the 20 ns coincidence window programmed in the hardware.


**Causal Isolation:**


During the opening time of aperture, photons traveled a distance of 140 m. We introduced 200 m of single-mode fibers between source and mirror as in Figure 2. This achieves isolation of detectors and crystal when separated by the fibers. In our table-top setup, we only considered fiber isolation. The detectors physically sat next to each other on our bread board. They were within 10 cm physical distance, but their fiber optical paths were separated by 400 m (2×200 m) of single-mode transport. Detector-to-detector optical isolation is a byproduct of source to detector fiber isolation as we double the distance of fiber transport needed to 400 m.


**Degradation:**


Luminosity and time of aperture are proportional. The shorter the time gate, the shorter the exposure to the light, and the shorter the collection of photons. This is true of single counts. The coincidence count is proportional to the single count, and not a quadratic of the single count. As we have observed, the mirror illuminates the separate individual slits at the same time; thus, the gate effects at distant slits are in sync. This is the reason we detect coincidences in the first place. This yields a linear relationship between coincidences and singles. The floor in our experimental setup was set by the dark count of singles and the dark count of coincidences. With rotation, the photons were spread over 2 lines that repeated 1000 times per second. We calculate the signal degradation as:(2)$$D_{th}=\frac{A\cdot N}{2\pi r}=0.016$$
with the aperture size $A=10^{-3}$ m, the radius mirror to slit r=0.34 m and the number of facets N=34.


**Source Characterization**


The singlet state coming out of the source was characterized by both QuTools and the author, using, respectively, a full tomography and a custom design involving a quarter wave plate in one arm, as
(3)$$|\Psi \rangle \approx 0.75(|H\rangle |H\rangle +0.88e^{-i0.28}|V\rangle |V\rangle )$$

With this source, and without mirror gating, we have routinely observed Bell–CHSH measures up to the S=2.70 range.


**Axes of Measurement**


The polarization measures were conducted along the usual angles of (0,45,90,135) for channel A and (22.5,67.5,112.5,157.5) for channel B.

## 3. Results


**Dark counts and detection window:**


Our APDs had individual dark count rates of 1300 s${}^{-1}$ and 600 s${}^{-1}$, and the dark count coincidence rate was approx 300 h${}^{-1}$. The time for the detection window of coincidences in APDs was 20 ns and was set in the hardware by quTools. We did not control this time window parameter. See Table 1.


**Degradation:**


The results of measurements for determining the luminosity and signal degradation due to the rotating Foucault mirror can be seen in Table 1. The dark counts of the detectors were measured in order to be subtracted from the results in further calculations.
(4)$$D_{ex}=\frac{\mathrm{Signal}\;\mathrm{with}\;\mathrm{rotation}}{\mathrm{Signal}\;\mathrm{without}\;\mathrm{rotation}}=0.018$$
The first observation is that we retain a clear coincidence signal, well above the noise level. The degraded coincidence signal in our experiments is on the order of 30,000 h${}^{-1}$, and the noise for said coincidences is measured at 300 h${}^{-1}$. We have a clear 100:1 contrast in the signal-to-noise ratio. The ratio between singles and coincidences is due to the quantum efficiency of the detectors. The higher value of the experimentally observed signal degradation could be due to the non-linear quantum efficiency of the detectors: the higher the measured count rate, the lower the quantum efficiency.


**Bell–CHSH observation:**


We have observed BIV of Bell–CHSH S=2.30±0.07. See Table 2.

## 4. Conclusions

We demonstrated that the AE/OMs Bell experiments close the locality loopholes between observers but do not causally isolate the sources from the observers in classic 4-channel loophole experiments. We introduced a spinning mirror to create such isolation in a 2-channel Bell experiment. By intermittently removing this hypothetical background, we anticipated a loss of the Bell effect (loss of BIV). We reported instead on observed Bell violations.

Regarding a hypothetical background mediating an influence from the polarizers to the emitting atoms and which is responsible for the Bell effect, we will now prove that the influence cannot be a traveling wave.

To prove this statement, we first observe that the time the gate is open is $4.7\times 10^{-7}$ s and that the distance covered by photons in the single-mode fibers during this time is approximately 140 m. Let us assume, for the sake of argument, that the influence from the detector to the source is instantaneous. The BBO crystal would start emitting entangled photons under the influence of the detector as soon as the gate opens. The photons informed by this instantaneous influence would then still need to cover the 200 m of single-mode fiber that separates them from the detectors. By the time the photons reach the mirror, the slit is not in view anymore. These photons are not detected. We then observe that this is also true of a slower influence signal: the influence will simply take longer to reach the source and delay the return trip by as much.

We therefore conclude that the influence cannot be that of a travelling wave, not even super-luminal, not even an instantaneous wave. This experimentally rules out all super-luminal theories explaining Bell. QED.

## 5. Discussion

For discussion, we offer the following logical argument. If a background field is responsible for Bell violations, then the communicating waves are either travelling or standing.

Either:1/ There is no background influence, or;2/ There is a background influence, and either:–2a/ It is a traveling local wave, or;–2b/ It is not (standing wave or deBroglie hypothesis).

The logical structure of the above is a list enumeration of the possible scenarios: ∄ background ∨(∃ background ∧( standing ∨¬standing)). This is declarative; there are no physics yet. The only physics contained in this statement is the hypothesis of a wave mediating the Bell effect, or not.

As for the physical content, one can argue in scenario 1 that there is no background field; it simply does not exist and something else is responsible for the Bell effect. This is the view espoused in the contextuality approach [14,15]: neither travelling nor standing background waves are needed in order to cause BIV. In this scenario, all we have done here is to reduce luminosity.

In scenario 2, we still consider the existence of a background influence. The conclusion of this paper was to experimentally rule out any travelling wave (including super-luminal) or option 2a.

Therefore, we are left with option 2b. The Bell effect cannot be due to traveling waves. What form could this wave be? Standing waves come to mind. The author looks to the hydrodynamic walkers for visual inspiration and intuition [21]. A droplet is bouncing and creating impact waves. In its wake are standing waves. The sum of these standing waves leads to a memory. This memory is a time-averaged wake, and the dynamics it leads to are chaotic [22]. These are the Faraday wave-fields, which are manifested in the hydrodynamic walkers [23]. In the hydrodynamic quantum analog field (HQA) of the walkers, the walkers follow a cavity geometry, which is represented by a ghostly but emerging mean field memory, which includes a form of echo-location. This mean field of standing waves represents a memory of past events, the wake, and usually includes information from the enclosing cavity, in the form of standing waves of the cavity, and this resonant memory guides the particles. The walkers demonstrate that by communicating via standing waves, a bipartite system of walkers behaves in a non-separable fashion, and [24] have used that to demonstrate a hydrodynamic analog of super-radiance. The standing waves represent a form of memory, and this memory ensures that the correlations between droplets is not communicated via instantaneous signaling (between either measurement stations or sources) but rather built up over time. Indeed, with hereditary systems, the notion of a light cone is all but irrelevant. This is also in alignment with the deBroglie’s intuition of matter modelled with standing waves about singular points.

## 6. Outlook and Prospects

As noted, the above experiment is performed on a commercially available 2-channel static measure setup. It is not a state of the art 4-channel dynamic-choice experiment for the simple reason that the author does not have access to such equipment (e.g., AEOMs). It has also been noted that the isolation between the source and the detectors is non-existent in the classic 4-channel experiments. When one considers the source-to-polarizer isolation, there is no loophole closing. The lines of sight and settings are all static. It would strengthen both the conclusions in this paper, as well as the loop-hole closing conclusions of the classic papers, if such a 4-channel dynamic-choice test were to be performed in the presence of this Foucault mirror gate. Based on the present positive result, the author would not expect a different outcome: i.e., we should still observe BIVs in such an enhanced geometry. The author is open to lending his mirror equipment to, and collaborating with, laboratories capable of conducting such experiments.

## Figures and Tables

**Figure 1 entropy-24-01230-f001:**
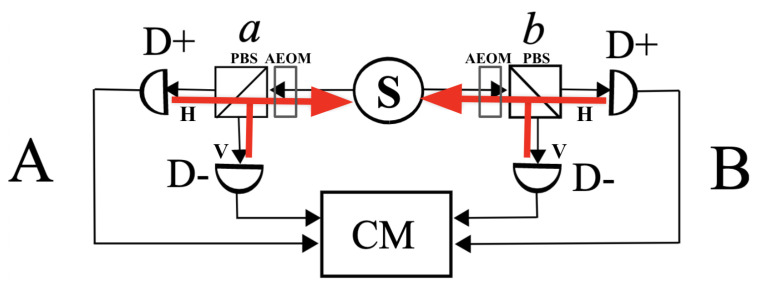
In red, the schematic representation of the static line of sight between analyzers (*a*, D+, D−) and source *S* on side (**A**) and similar on side (**B**) (*b*, D+, D−). Components *a* and *b* represent polarizing beamsplitters fronted by modulators that are either AOM or EOM actuators. Component CM is the coincidence measuring apparatus. There is no line of sight between Alice and Bob. However, there are lines of sight between the source and the four detectors. The source always sees a superposed signal (has a static line of sight) from all 4 detectors with their polarizations at all times.

**Figure 2 entropy-24-01230-f002:**
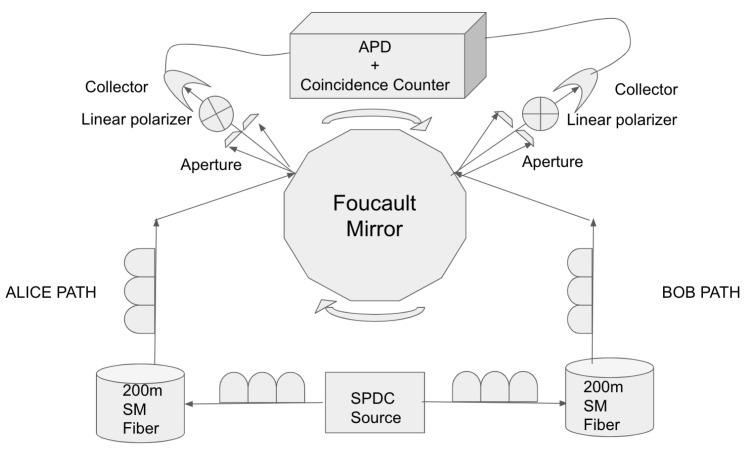
Schematic layout. The spontaneous parametric down conversion (SPDC) source was down in the middle. Alice and Bob photons were transported over a 200 m single-mode fiber. We compensated with 3× bat ear wave delays both at the input and output of both fibers for a total of 4 bat ears. Both paths were bounced off of the Foucault mirror, which created a line arc over an aperture at the detectors. We then filtered through static computerized polarizers into collectors and into the avalanche photodetectors (APD). Finally a coincidence counter from quTools detected coincidences in channel A and B within a 20 ns window.

**Table 1 entropy-24-01230-t001:** Results of signal luminosity and degradation measurements, in photons per second.

	Singles A (/s)	Singles B (/s)	Coincidences (/s)
Dark Counts	1300	600	0.08
No rotation	33,894	20,329	389
With rotation	2301	1098	7
Degradation $D_{ex}$	0.031±0.003	0.025±0.004	0.018±0.008

**Table 2 entropy-24-01230-t002:** CHSH S=2.302±0.071>2 . There are 16 measurement settings for the polarizers. In blue, one can see the individual counts, and in grey to the right, one can see the accidental coincidence counts (noise because of single counts). For example, the first cell is 0:22.5, the coincidence count is 226 photons and the noise is 5 photons. The integration time is 60,000 ms (1 min per measure, 16 measures, 16 min total). The 1-min per-measure setting is the maximum setting we can program within the quTools environment.

Angle in ${}^{\circ }$	0	45	90	135
22.5	226 -5	85 -4	42 -4	184 -4
67.5	70 -5	34 -4	182 -4	239 -5
112.5	46 -5	187 -5	227 -4	70 -4
157.5	198 -5	217 -5	88 -4	34 -4

## Data Availability

Not applicable.

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
