# Peer review of "Observations of Bell Inequality Violations with Causal Isolation between Source and Detectors"

_entropy, 2022, doi:10.3390/e24091230_

Round 1
Reviewer 1 Report
See attached review.

Author Response
*All* points by Reviewer 1 have been addressed:
Specifically:
1/ Correct reference to Vervoort has been included. This was an oversight on my part for it was the work I simulated in the past and which built my intuition
2/ Reference to the hydrodynamic analog of superradiance is indeed very relevant and has been included almost as-is in the Discussion part.
3/ We have put more emphasis on memory as a key ontological ingredient
On the writing feedback
line 21: ‘ ...we will motivate...’ → ‘ ...we will argue...’ • line 22. ‘It follows that we will lose BIV by removing the line of sight between source and detection’. It would be more accurate to replace “will” with “might” • the CHSH acronym should be defined the first time it appears in the text • line 40. ‘the original experiments by A.A.’ : I presume A.A stands for Alain Aspect, which should be specified • Figures 1 & 2 captions: Start with a capital letter. 2 • BBO crystal should be defined. . •
All fixed
line 65:: [9] considers [8] wrong claiming it implies a super-luminal communication in the background. Standard writing would be something along the lines of: “Schmeltzer [9] contests the conclusion of Vervoort [8] on the grounds that..... Note also that in his 3-page article, Schmeltzer [9] does not show any mathematical errors in the model of Vervoort [8]; rather, he rejects the model premises from the start, affirming without argument that the model is non-local. This amounts to saying: this model contradicts the standard view, so it must be wrong. Schmeltzer has written similarly light-weight articles declaring his unfounded disagreement without substantive criticism about other works, including Ref. [12] by T. Nieuwenhuizen. I leave it to the author whether he thinks such declarations are worth citing.
I have removed the Schmeltzer citation
• line 111. Explanation of the experimental setup is too brief for a scientific paper. Please elaborate in the text or show more detailed schematics. You should provide enough technical detail for the interested reader to be able to reproduce your experiments. Details can be placed in a Supplementary Information section.
I have beefed up the description to include every component
Figure 2 caption: ‘ are transported over A 200m...’. Also should it not be ‘...input and output of both fibers..’? Finaly, I am not familiar with the meaning of (1/4.1/2.1/4) wave dealys. This sentence needs rewriting. • line 119. the notation of the distance R in line 119 is very unconventional. Please consider changing it. • line 120: ‘alight’ → ‘illuminated’ • line 126: Should be ‘During the opening time of the aperture...’ • line 133. ‘This yields a linear relationship between coincidences and singles, mainly due to the quantum efficiency of the detectors.’ I do not understand this sentence. Please clarify.. • In Table 1, please clarify what is meant by: Single 0 (/s) Single 1 (/s) Coinc. (/s)
All fixed
• line 175: I think that plain english would be more clear than the language of logic adopted at the beginning of the Discussion
I have included a plain english version of it but keep the logical argument in logical format for completeness and rigor.
Reviewer 2 Report
Through an extensive literature review, the author derived a research question on whether the EM background field between observers and emitters in the Bell tests affects the outcomes. He negatively answered this question with a two-channel Bell test using a rotating Foucault mirror.
In my view, his research question is interesting since its experimental verification could be relevant for the delimitation of hidden variable theories to be excluded by the Bell tests in detail. However, it seems to me that the experiment given in this paper is too weak to support his conclusion. More concretely, the following points should be improved:
1. Section 1.2 seems too concise to convey the idea. This is because Figure 1 has no information about a line of sight. This makes it difficult to imagine the experimental configuration criticized in this paper. The author should replace the figure with a more detailed one.
2. It is unclear whether the present experiment is locality-loophole-free concerning the stations. The author should append some comments in the manuscript.
3. It is unclear whether the present experiment implements active choices of the measurement axes in each station.
4. It is desirable to give an account of why S=2.30 by introducing descriptions of the initial state and the directions of the measurement axes.
Bearing these points in mind, the present manuscript should be evaluated as a note of preliminary work, which has the potential to be grown to a full-scale experiment. Thus, I would like to reject the present manuscript for now and allow the author to choose the direction of how to revise this manuscript.
Author Response
Reviewer 2.
Point 1. Section 1.2 seems too concise to convey the idea. This is because Figure 1 has no information about a line of sight. This makes it difficult to imagine the experimental configuration criticized in this paper. The author should replace the figure with a more detailed one.
Answer: Figure replaced. Using the original layout for a clearer classic map. I have also updated the caption to further clarify the line of sight.
Point 2: It is unclear whether the present experiment is locality-loophole-free concerning the stations. The author should append some comments in the manuscript.
Answer: we have updated section 2.1 to explicitly say that we do not implement a locality loophole-free test in the usual sense of Aspect and Zeilinger since the polarizers are physically close to each other on the breadboard. The isolation considered here is only between the polarizers and the source and via the fiber. The polarizers and the source are also physically close on the breadboard. It is the optical path of the photons that is locally separated by the fiber and the mirror
Point 3. It is unclear whether the present experiment implements active choices of the measurement axes in each station.
Answer: It does not, and I have updated section 2.1 to reflect this fact explicitly
Point 4. It is desirable to give an account of why S=2.30 by introducing descriptions of the initial state and the directions of the measurement axes.
Answer Ψ = (1+A2)-1/2 (|0>a |0>b + Aeiφ|π/2>a |π/2>b ) with A =0.88 and phi=-.28. This is reflected in subsection 2.1 paragraph Source characterization
The axes of the measures were reported in table 2. But I have also explicitly updated subsection 2.1 paragraph axes of measurement.
Bearing these points in mind, the present manuscript should be evaluated as a note of preliminary work, which has the potential to be grown to a full-scale experiment.
The points about a full-blown experiment are duly noted and I do agree with reviewer 2 that a classic 4-channel with loophole closing might be relevant. However, I only have access to a 2-channel experiment and not to the more sophisticated dynamic choice 4 channel in the vein of Weihs and Zeilinger. For what it is worth I do not expect a different result even in the 4-channel experiments and would not recommend pursuing it.
Reviewer 3 Report
The present work proposes an experimental observation of violations of Bell’s inequality with causal isolation between source and detectors. The author, making use of Foucault's mirror gating geometry, asses to eliminate the causal connection between entangled photon source and detector setups. This paper attempts to answer some questions that have arisen recently. In Ref. [8] of the main text, the authors verified the violation of CHSH even assuming the existence of a de-localized background field whose existence breaks the measurement independence assumption. Therefore, the question the author want to solve is, ”Can a de-localized background field be the cause of a violation of Bell’s inequality?”. In the present work, the
violation of CHSH is observed with a reduced value even by ”eliminating the line between source and detection.” Therefore, the author concludes that there cannot be a background influence due to a local traveling wave.
In my opinion, it is not clear which model the experiment wants to exclude.
in fact, in the manuscript the source and detectors are considered isolated, in the sense that the gate opening time is less than the time that a photon generated under the influence of the measurement result takes to reach the Foucault mirror and, therefore, this photon is not detected. But it is not clear to me why this implies that the influence cannot be that of a traveling wave. Are you considering a wave traveling at the maximum speed of light and
following exactly the same trajectory as photons? Also, the CHSH is evaluated by collecting photons in 1 minute per measurement that is longer than the gate opening time, so how should this prevent influence due to traveling waves? Moreover, since the locality loophole is not closed a lot of different models can still be considered, so I think this work is not relevant and, for all these reasons, I do not recommend its publication.
However, I have some suggestions that could help the author in future submissions. First, the exposition could be improved. The writing is not very smooth and the theoretical background on Bell inequalities and quantum correlations is completely lacking. Also, the bibliography is mainly limited to the exposition of the central issue of the article, but no recent and historical articles, such as those concerning the EPR paradox or Bell’s inequalities are cited. The author states that ”By intermittently removing this hypothetical background we anticipated a loss of the Bell effect (loss of BVI)”, but no comparison values are provided. In particular, the violation of Bell’s inequality without the use of the Foucault's mirror should be reported. Also, I think the author should clearly report all the loopholes still open in the experiment and the specific model he wants to rule out.
Author Response
Reviewer 3. Paragraph 2. Question 1
Question: In my opinion, it is not clear which model the experiment wants to exclude. in fact, in the manuscript the source and detectors are considered isolated, in the sense that the gate opening time is less than the time that a photon generated under the influence of the measurement result takes to reach the Foucault mirror and, therefore, this photon is not detected. But it is not clear to me why this implies that the influence cannot be that of a traveling wave. Are you considering a wave traveling at the maximum speed of light and following exactly the same trajectory as photons?
Answer: Yes we only consider the wave traveling along the same trajectory as the photons. This is to say that we only concern ourselves with signals carried by the fibers. However, we do not only rule out fiber waves traveling at a maximum speed of light but also those traveling at super-luminal (including instantaneous) speeds. We do not concern ourselves with signals not carried by the fiber and specify that the physical dimensions of the breadboard are 1mx1m.
Reviewer 3. Paragraph2. Question 2
Question: Also, the CHSH is evaluated by collecting photons in 1 minute per measurement that is longer than the gate opening time, so how should this prevent influence due to traveling waves?
Answer: I must admit to being a little confused by this question. The settings are all static this is true. The Avalanche Photo Detector (APD) is always on, and the Polarizers are set every minute. Data collection lasts a minute. However, the Foucault mirror chops the line of sight between the polarizers and the source (so a fortiori the APD and the source since the APDs sit behind the polarizers). The EM line of sight is on only for 5x10^-7s, and photons can travel 140m during that time. Any EM signal is not present and destroyed by the closed gate. Assume the line of sight opens up and an influence from the polarizers or APD travels to the source, then the photons produced under the influence cannot reach the polarizer for they need to travel 200m and the gate is closed after 140ms. This prevents the photons under EM influence from ever being detected.
Round 2
Reviewer 2 Report
I am glad to find that the author of the present paper dealt with all the things I pointed out.
His revision improved the presentation extensively, resulting in the elucidation of both the limitation and contribution of this paper.
However, although I also believe that the 4-channel active-choice Bell test could yield the same results, I think that the author should append some comments on the necessity of such experiments to strengthen the present result. It is because someone (not necessarily the present author) interested in this paper could try to such a test.
It will be good to do so at the end of the discussion section, by making a paragraph concerning outlook or prospects.
Best
Author Response
I have added a paragraph called "Outlook and Prospects" at the end of the discussion.
Reviewer 3 Report
The author followed almost all of my suggestions in the revised version, but I think this work is still not that interesting. It focuses on rejecting an extremely specific condition in which there could be a background field that follows exactly the same trajectory as the photons. However, many different models can be considered, for example, the background field might not be an electromagnetic field and therefore not follows the same trajectory as photons, or it might be an electromagnetic field with a different wavelength that is transmitted by tilted mirrors. Also, in my opinion, if the polarizer settings are changed every minute, the considered model is not properly ruled out, since each measurement should be shorter than 5x10 ^ -7s, and more statistics should be collected by repeating each short measurement. Without closing the locality loophole, many different models can still be considered; for all these reasons, I do not recommend publication.
Author Response
Opinions noted.